# Conjugative RP4 Plasmid-Mediated Transfer of Antibiotic Resistance Genes to Commensal and Multidrug-Resistant Enteric Bacteria In Vitro

**DOI:** 10.3390/microorganisms11010193

**Published:** 2023-01-12

**Authors:** Azam A. Sher, Mia E. VanAllen, Husnain Ahmed, Charles Whitehead-Tillery, Sonia Rafique, Julia A. Bell, Lixin Zhang, Linda S. Mansfield

**Affiliations:** 1Comparative Enteric Diseases Laboratory, East Lansing, MI 48824, USA; 2Comparative Medicine and Integrative Biology Graduate Program, College of Veterinary Medicine, Michigan State University, East Lansing, MI 48824, USA; 3BEACON Center for the Study of Evolution in Action, Michigan State University, East Lansing, MI 48824, USA; 4Department of Microbiology and Molecular Genetics, Michigan State University, East Lansing, MI 48824, USA; 5Department of Large Animal Clinical Sciences, Michigan State University, East Lansing, MI 48824, USA; 6Department of Epidemiology and Biostatistics, Michigan State University, East Lansing, MI 48824, USA

**Keywords:** antibiotic resistance, RP4 conjugative plasmid, enteric pathogens, commensal bacteria

## Abstract

Many antibiotic-resistant bacteria carry resistance genes on conjugative plasmids that are transferable to commensals and pathogens. We determined the ability of multiple enteric bacteria to acquire and retransfer a broad-host-range plasmid RP4. We used human-derived commensal *Escherichia coli* LM715-1 carrying a chromosomal red fluorescent protein gene and green fluorescent protein (GFP)-labeled broad-host-range RP4 plasmid with *amp*R, *tet*R, and *kan*R in in vitro matings to rifampicin-resistant recipients, including *Escherichia coli* MG1655, Dec5α, *Vibrio cholerae*, *Pseudomonas putida*, *Pseudomonas aeruginosa*, *Klebsiella pneumoniae*, *Citrobacter rodentium*, and *Salmonella* Typhimurium. Transconjugants were quantified on selective media and confirmed using fluorescence microscopy and PCR for the GFP gene. The plasmid was transferred from *E. coli* LM715-1 to all tested recipients except *P. aeruginosa*. Transfer frequencies differed between specific donor–recipient pairings (10^−2^ to 10^−8^). Secondary retransfer of plasmid from transconjugants to *E. coli* LM715-1 occurred at frequencies from 10^−2^ to 10^−7^. A serial passage plasmid persistence assay showed plasmid loss over time in the absence of antibiotics, indicating that the plasmid imposed a fitness cost to its host, although some plasmid-bearing cells persisted for at least ten transfers. Thus, the RP4 plasmid can transfer to multiple clinically relevant bacterial species without antibiotic selection pressure.

## 1. Introduction

Antimicrobial-resistant pathogens are a leading cause of death worldwide. A recent study estimated that approximately 4.95 million deaths occurred globally due to drug-resistant bacteria in 2019 [1]. Similarly, the Centers for Disease Control and Prevention (CDC) reported that antibiotic-resistant (AR) pathogens in the USA caused ~3 million infections and 36 thousand deaths during 2019 [2]. If antibiotic resistance concerns are not adequately addressed in the near future, it has been estimated that AR bacteria will cause significantly more deaths: as many as 10 million each year globally, posing a significant economic burden of up to USD 100 trillion by 2050 [3]. These findings highlight the fact that the emergence of AR pathogens has become a major global health challenge.

Broad-host-range (BHR) conjugative plasmids capable of transferring to a wide range of closely and distantly related bacteria are the leading cause of ARG spread among multiple species of bacteria [4,5,6,7] and are divided into different incompatibility groups based on their stability during the conjugation process [8,9]. BHR plasmids associated with the IncP group, especially the RP4 plasmid (IncP-1α), have shown massive potential for spreading and transferring ARGs to new bacterial hosts [10,11,12]. The RP4 plasmid (IncP-1α) isolated from human clinical samples has been one of the most highly studied plasmids since its discovery in the 1970s [13,14]. The RP4 plasmid has various genetic features to maintain self-replication, transmissibility, and survival in a wide range of Gram-negative and Gram-positive bacteria [15,16]. Previous studies have shown the transfer of the RP4 plasmid from laboratory strains of *E. coli* to bacteria isolated from soil [11,17,18], sewage, and activated sludge [19,20,21]. In a recent study, Heß et al. (2022) reported that in microcosm experiments, RP4 plasmid transferred to multiple bacterial strains from three donor *E. coli* strains [22]. These past studies on RP4 conjugation were mainly conducted using laboratory or environmental donor *E. coli* strains. However, a major knowledge gap exists in understanding the transfer of a broad-host-range plasmid such as the RP4 to other bacteria of the human gut microbiota.

Most of the antibiotic-resistant bacteria identified to date are either enteric pathogens or capable of residing in the human gut [1,2]. Bacterial species belonging to families *Enterobacteriaceae*, *Pseudomonadaceae*, and *Vibrionaceae* are among the leading causes of antibiotic-resistant infections worldwide [1]. Compared to other bacterial families, members of *Enterobacteriaceae* serve as hosts for the greatest number of antibiotic-resistance-encoding plasmids [23] including both broad- and narrow-host-range plasmids (extensively reviewed by Carattoli (2009) and Rozwandowicz et al. (2018)) [9,24]. To date, 28 different incompatibility (Inc) groups of plasmids in *Enterobacteriaceae* and 14 in *Pseudomonadaceae* have been reported, including the RP4 plasmid [23]. However, little is known about the transfer frequencies and persistence of these BHR plasmids among enteric bacteria belonging to the families *Enterobacteriaceae*, *Vibrionaceae*, and *Pseudomonadaceae*. These clinically relevant AR bacteria live in the human gut, a complex environment where they have great potential to share resistance-gene-bearing plasmids with other commensal and pathogenic bacteria and to convert recipients into potential emerging drug-resistant bacteria. Furthermore, plasmid acquisition imposes a fitness cost on recipient bacteria that determines plasmid persistence in a complex microbial community [25,26]. Thus, it is difficult to develop strategies to mitigate the spread of ARGs without understanding the frequency of spread and the persistence of BHR plasmids among these clinically relevant bacteria. Therefore, we investigated RP4 plasmid transfer from a commensal human gut *E. coli* bacterium to other closely and distantly related enteric bacteria, which have not been examined previously.

In this study, we determined the potential for primary spread of a broad-host-range plasmid RP4 from commensal gut *E. coli* to multiple naïve host bacteria of the gut microbiota and, once transferred, subsequent secondary transfer from those hosts back to plasmid-free commensal *E. coli* (Figure 1). We hypothesized that (i) the RP4 plasmid would transfer to many species of *Gammaproteobacteria* in the absence of antibiotic treatment, (ii) the different pairings of donor and recipient strains would influence plasmid transfer frequency, and (iii) the RP4 plasmid can persist in a naïve host without antibiotic selection pressure. To address these hypotheses, we chose a sample set of clinically relevant bacterial species of the *Enterobacteriaceae*, *Vibrionaceae*, and *Pseudomonadaceae* families to act as RP4 donors and recipients. In these experimental transfers, we used a commensal *E. coli* LM715-1 originally isolated from a human infant gut microbiota as the primary donor and recipient strain. RP4 plasmid persistence in this strain was measured using a serial transfer experiment. Overall, our study showed that the (i) the RP4 plasmid could transfer to multiple bacterial species efficiently, (ii) the specific donor–recipient pairing affected the transfer frequency of the plasmid, and (iii) the RP4 plasmid imposed a fitness cost on a naïve host but persisted in a small proportion of the population.

## 2. Materials and Methods

### 2.1. Media, Chemicals, and Reagents

Luria agar (Acumedia, Lansing, MI, USA), LB-Miller broth (IBI Scientific, Dubuque, IA, USA), MacConkey agar (Neogen, Lansing, MI, USA), and bacteriological agar (Neogen) were used to grow donor, recipient, and transconjugant bacteria. Bacterial culture experiments for *Salmonella enterica* serovar Typhimurium, *Citrobacter rodentium* ATCC 51459, *Klebsiella pneumoniae* IA565, *Vibrio cholerae* O1 biotype El Tor C6706str2, *Pseudomonas aeruginosa*, *Pseudomonas putida* KT2440, *Escherichia coli* LM715-1, *Escherichia coli* MG1655, and *Escherichia coli* DEC 5a TW00587 were performed aerobically at 37 °C either in an incubator (plates) or on a shaker at 150 rpm (liquid cultures). We used antibiotics throughout the study as required for specific selection at the following concentrations: ampicillin, 50 μg/mL; kanamycin, 50 μg/mL; chloramphenicol, 20 μg/mL; tetracycline, 15 μg/mL; rifampicin, 20 μg/mL; and cefotaxime, 4 μg/mL. We used antibiotic combinations with the same concentrations to select donor, recipient, and transconjugant bacteria, depending on the donor and recipient bacterial strains used in a specific experiment. The use of specific antibiotics is documented for each experiment in the Results section. We prepared phosphate-buffered saline (1X PBS) using the following recipe: NaCl: 137 mM, KCl: 2.7 mM, Na2HPO4: 4.3 mM, and KH2PO4: 1.4 mM). The sources of antibiotics were ampicillin (Millipore Sigma, Burlington, MA, USA), kanamycin (Fischer Scientific, Fairlawn, NJ), chloramphenicol (M.P. Biomedicals, Solon, OH, USA), tetracycline (M.P. Biomedicals, Solon, OH), rifampicin (Alfa Aesar, Tewksbury, MA, USA), and cefotaxime (Alfa Aesar, Tewksbury, MA, USA).

### 2.2. Isolation and Characterization of a Human-Derived Donor Bacterial Strain

Human-derived, mouse-adapted commensal *E. coli* LM715-1 was isolated from a mouse that carried a human infant fecal microbiota transplanted to germ-free mice and passaged vertically to subsequent generations in the Mansfield laboratory colony (Moya and Mansfield, unpublished). This is a Biosafety Level 2 colony managed to prevent any acquisition of bacteria extraneous to the defined human source [27]. These mice were transplanted with fecal slurries of children recruited into the Isle of Wight “Third Generation Study” under UK ethics approval numbers 09/H0504/129 (22 December 2019), 14/SC/0133 (22 December 2019), and 14/SC/1191 (15 November 2016).

A DNeasy blood and tissue kit (QIAGEN, Catalogue. No./ID: 69504, Germantown, MD) was used to extract genomic DNA of an *E. coli* colony isolated from this mouse. Next, we performed multilocus sequence typing (MLST) of the *E. coli* LM715-1 strain using a scheme of seven housekeeping genes: *aspC*, *clpX*, *fadD*, *icdA*, *lysP*, *mdh*, and *uidA*. This MLST scheme can characterize pathogenic *E. coli* isolates using an existing database of the seven abovementioned housekeeping genes. The primers and protocols for MLST available on the website (http://shigatox.net/ecmlst/cgi-bin/scheme, accessed on 11 October 2019) were used to amplify these genes. Sanger sequencing of these amplicons was performed at Michigan State University Research Technology Support Facility Genomics Core. Sequence analysis was performed using an online tool available on the database website (http://shigatox.net/ecmlst/cgi-bin/dbquery, accessed on 30 March 2020) of Shiga-toxin-producing *E. coli* (STEC).

We also submitted DNA of *E. coli* LM715-1 for whole-genome sequencing at the Michigan State University Research Technology Support Facility Genomics Core. Sequencing was carried out in a 2 × 250 bp paired-end format using a MiSeq v2 500 cycle reagent cartridge. Base calling was performed using Illumina Real Time Analysis (RTA) v1.18.54, and the output of RTA was demultiplexed and converted to FastQ format with Illumina Bcl2fastq v2.20.0. A FastQC report of the run output was created to determine the quality scores of the sequenced data. The sequence dataset can be accessed at the Dryad database with this identifier number: doi:10.5061/dryad.v41ns1s16 (https://datadryad.org, accessed on 2 December 2022). Next, sequences were analyzed using EnteroBase, an online integrated software platform, to perform MLST on commensal human gut *E. coli* LM715-1 using the Achtman MLST scheme [28]. Seven loci (*adk*, *fumC*, *gyrB*, *icd*, *mdh*, *purA*, and *recA*) were used for typing of commensal human gut *E. coli* LM715-1 [29]. The Bacterial and Viral Bioinformatics Resource Center (BV-BRC) (https://www.bv-brc.org/, accessed on 15 November 2022), a web-based interface, was used to assemble, annotate, and analyze the commensal human gut *E. coli* LM715-1.

To determine antibiotic resistance phenotypes of this *E. coli* LM715-1 isolate that could be employed as selective markers for conjugation experiments, we performed antibiotic resistance profiling of this *E. coli* isolate against the following antibiotics at specific concentrations: ampicillin (50 μg/mL), kanamycin (50 μg/mL), chloramphenicol (20 μg/mL), tetracycline (15 μg/mL), rifampicin (20 μg/mL), and ceftriaxone (4 μg/mL). These specific antibiotic concentrations were selected based on the resistance markers present on the RP4 plasmid, donor, and recipient bacteria and used throughout the study.

### 2.3. Creation of a Fluorescently Labeled Commensal Donor Strain

To create a trackable donor strain, we obtained a chromosomal insertion toolbox designed by Schlechter et al. (2018) [6] from the Addgene plasmid repository (Watertown, MA, USA) (https://www.addgene.org, accessed on 5 November 2019), which consists of a Tn7-based thermo-unstable delivery suicide plasmid (pMRE-Tn7-155 plasmid, Addgene Plasmid #118569). We inserted a genomic cassette (mScarlet fluorescent protein gene, KanR, and CamR) into the *E. coli* LM715-1 bacterial chromosome using a pMRE-Tn7-155 delivery plasmid following the method described by Schlechter et al. (2019) [30]. The insertion and activity of the fluorescence gene and antibiotic resistance marker genes were confirmed using fluorescent microscopy, selection on LB agar with chloramphenicol (20 μg/mL) and kanamycin (50 μg/mL), and colony PCR (see below) for the mScarlet fluorescent marker gene.

*E. coli* MG1655 bearing a broad host range (BHR) RP4 plasmid labeled with a green fluorescent protein marker and antibiotic resistance genes *amp*R, *tet*R, and *kan*R was the kind gift of Dr. Barth Smets’ research group at the Technical University of Denmark [11]. We mated the fluorescently labeled commensal strain *E. coli* LM715-1 (mScarlet, *cam*R, *kan*R) as a recipient strain with donor *E. coli* MG1655 (RP4::GFP, *amp*R, *tet*R, and *kan*R) using the filter conjugation method described below. A single colony from the transconjugant selective medium (Luria agar containing chloramphenicol and ampicillin) was picked and streaked on a fresh Luria agar plate containing chloramphenicol and ampicillin, and plates were incubated aerobically at 37 °C overnight. After streaking twice for purity from single colonies, a bacterial lawn of *E. coli* LM715-1 carrying the RP4 plasmid was suspended in LB-Miller broth containing 30% glycerol and stored at −80 °C. The donor strain was restreaked from this stock culture for all experiments.

### 2.4. Donor and Recipient Strains

In this study, we examined the transfer of the broad-host-range RP4 plasmid in clinically relevant human-derived enteric commensal and pathogen strains. We chose a diverse collection of recipient strains, including *Salmonella enterica* serovar Typhimurium, *Citrobacter rodentium* ATCC 51459, *Klebsiella pneumoniae* IA565, *Vibrio cholerae* O1 biotype El Tor C6706str2, *Pseudomonas aeruginosa*, *Pseudomonas putida* KT2440, commensal human gut *Escherichia coli* LM715-1, Diarrheagenic *Escherichia coli* DEC 5a TW00587, and K-12-derived *Escherichia coli* MG1655 (Table 1). Three different strains of *E. coli* were included to determine differences in plasmid transfer frequencies within a single bacterial species. Diarrheagenic *E. coli* DEC 5a belongs to the sequence type ST-73, *and* commensal *E. coli* LM715-1 belongs to STEC Center MLST-defined sequence type ST 259 and *K-12-derived E. coli* MG1655. To identify antibiotic resistance markers for conjugation assays, we conducted antibiotic resistance profiling of these strains by growing them in Luria broth with a panel of antibiotics in the concentrations specified above. We found that none of the bacteria were resistant to rifampicin. Therefore, we selected for rifampicin-resistant spontaneous mutants as a means of positively identifying RP4 recipients. We isolated rifampicin-resistant spontaneous mutants of all recipient bacteria by streaking on Luria agar plates containing rifampicin (20 μg/mL) and incubating overnight at 37 °C aerobically. Single mutant colonies were transferred to another Luria agar plate containing rifampicin (20 μg/mL) and incubated again overnight at 37 °C aerobically. After the third repetition of streaking of single mutant colonies on the Luria agar plate with rifampicin (20 μg/mL), these rifampicin-resistant colonies were stored in LB-Miller broth containing 30% glycerol at −80 °C. We used rifampicin only when no other usable antibiotic resistance marker was present on the recipient bacteria (Table 1). Before every conjugation experiment, we grew donor and recipient bacteria fresh from freezer stocks to confirm the presence or absence of antibiotic resistance markers as required for the experiment. Confounding spontaneous resistant mutations, such as RifR in the donor population, were detected but very rare. To prevent such problems, we also assessed the antibiotic resistance of selected colonies of donor, recipient, and transconjugant bacteria using LB agar plates with the combinations of different antibiotics used in the experiment.

### 2.5. In Vitro Conjugation Experiments to Confirm Plasmid Transfer

The plasmid transfer frequencies for all pairs of donor and recipient bacteria were assessed using a filter-based conjugation method [36] (Trieu-Cuot and Courvalin, 1985). Briefly, both donor and recipient bacteria were grown overnight in Luria broth with respective antibiotics as indicated in the experimental design at 37 °C and 150 rpm. Then, 750 μL of each donor and recipient culture were mixed, and a pellet was obtained after washing twice with phosphate-buffered saline (1X PBS) at 10,000× *g* for 5 min (Eppendorf™ centrifuge 5415D, F-45-24-11 rotor, Hamburg, Germany). After resuspending the pellet in 100 μL of PBS, 20 μL of suspension was spread on each of four to five separate filters on a Luria agar plate with or without ampicillin (50 μg/mL). Cellulose filter papers (Whatman catalog# 1001-125, Maidstone, UK) were cut with sterile scissors into small pieces of 2 cm × 2 cm. Filters were placed on Luria agar plates with or without ampicillin (50 μg/mL). After incubating plates at 37 °C overnight, each filter was placed in a 1.5 mL microcentrifuge tube, 1000 μL of PBS was added, and the tube was vortexed for 60 seconds before serially diluting the suspension. We spread 100 μL of these dilutions on each of three separate Luria agar plates with antibiotics selecting separately for the donor, recipient, and transconjugants. Plates were incubated aerobically overnight at 37 °C; colonies were counted to calculate plasmid transfer frequency using the formula $f=\frac{T}{R+T}$ (plasmid transfer frequency = transconjugants/(transconjugants + recipients)).

To test transfer of the marked RP4 plasmid from commensal gut *E. coli* LM715-1 to freshly obtained fecal coliform isolates from the transplanted mice with the infant fecal microbiota described above, we made slurries of fecal material from transplanted mice by suspending 400 μL of TSB with 20% glycerol, vortexing, and mixing with sterile wooden sticks. These fecal slurries were streaked separately on MacConkey agar containing antibiotics (kanamycin or ampicillin) to screen for resistance to plasmid markers kanamycin and ampicillin. After confirming the lack of antibiotic resistance to kanamycin and ampicillin, these fecal slurries were streaked on MacConkey agar, and plates were incubated at 37 °C aerobically. After overnight incubation, bacterial populations from these plates were suspended in Luria broth with 30% glycerol and stored at −80 °C. Next, we performed conjugation between these populations of coliform bacteria harvested from MacConkey agar and our fluorescently labeled donor strain of *E. coli* LM715-1+ RP4 using the filter-based mating conjugation protocol described above. For transconjugant selection, we performed a two-step screening, as our recipient bacteria were susceptible to antibiotic resistance markers on both the donor *E. coli* chromosome (*cam*R and *kan*R) and RP4 plasmid (*amp*R, *tet*R, and *kan*R). In the first step, we grew transconjugants on MacConkey agar plates containing ampicillin and kanamycin, which excluded all cells except transconjugant and donor bacteria. In the second step, we randomly labeled and picked individual colonies (~200 colonies) with a sterile toothpick and transferred them to MacConkey agar plates containing chloramphenicol. The bacterial colonies not growing on MacConkey agar plates with chloramphenicol were marked on the first plate as potential transconjugant bacteria. These transconjugant colonies were grown further on MacConkey plates containing ampicillin and kanamycin, saved in the Luria broth with 30% glycerol, and stored at −80 °C for further characterization. Matrix-assisted laser desorption ionization-time-of-flight mass spectrometry (MALDI-TOF) was used to identify the transconjugant bacteria [37]. Fluorescent microscopy and colony PCR, as described below, were performed to confirm the presence of plasmid in the transconjugant bacteria.

### 2.6. Detection of Fluorescence in Donor and Transconjugant Bacteria

Bacterial cells from donor and transconjugant selective media plates were suspended in PBS and centrifuged at 10,000× *g* for 1 min at room temperature in an Eppendorf™ 5415D centrifuge with an F-45-24-11 rotor (Eppendorf™, Hamburg, Germany). The pellet was resuspended in 100 μL of PBS. We used the agarose pad method to obtain images of fluorescent bacteria (donor and transconjugants) [38]. Briefly, a 1% agarose solution was poured on a plain surface bordered with microscope slides to achieve agar pads of even thickness. After the agarose solidified, coverslip-sized pads were cut using a sterilized scalpel and placed on another microscope slide. A volume of 2–5 μL of bacteria suspended in PBS was spread over the agar pad and covered with a coverslip. A Nikon Eclipse N*i*-U upright microscope (Nikon, Tokyo, Japan) was used with bright-field, GFP, and RFP filters to record and analyze the fluorescent bacteria at magnifications of 20× and 40×. For rapid screening of fluorescent bacterial colonies, we took a small number of bacteria directly from the individual colonies grown on the plate using a sterile toothpick and mixed them with 10 μL of PBS or deionized water on a microscope slide. After placing a coverslip and air drying the slide for ten minutes, we observed fluorescent mScarlet and GFP expression in the donors, recipients, and transconjugants under the microscope.

### 2.7. Confirmation of the Plasmid in Transconjugant Bacteria Using Colony PCR

Colony PCR was performed to confirm the presence of RP4 plasmid carriage by the transconjugant bacteria using primers for the GFP marker located on the plasmid (Table 2). Colonies from transconjugant selective media plates were sampled using a sterilized toothpick and mixed with a 25 μL reaction mixture in a PCR tube. Each reaction mixture contained 2.5 μL 10× buffer (MgCl_2_ free), 2.5 μL MgCl_2_ (50 mM), 2.0 μL dNTPs (2.5 mM), 0.25 μL Taq DNA polymerase (New England BioLabs, Woburn, MA, USA), and 1.0 μL both forward and reverse primers (25 pM/μL). The final volume was adjusted with sterile distilled water up to 25 μL. DNA amplification was performed in a thermocycler (Eppendorf™, Model # AG 22331, Hamburg, Germany) using an initial denaturation step at 95 °C for 10 min followed by 30 cycles of amplification (denaturation at 95 °C for 1 min, annealing at 55 °C for 1 min, and extension at 72 °C for 1.5 min), ending with a final extension at 72 °C for 5 min. The PCR product was visualized by agarose (1.5%) gel electrophoresis to confirm the predicted 181-base-pair band for GFP present in the transconjugant colonies.

### 2.8. Bacteriocin Assay

A cross-streaking method was used to determine the bacteria-killing effect [39]. We streaked a Luria agar plate with *P. aeruginosa* using a sterilized cotton swab from fresh overnight culture and incubated the streaked plate for 24 h. The next day, we cross-streaked the plate with indicator strain *E. coli* LM715-1 using a sterilized cotton swab from fresh overnight culture and incubated it for 24 h. The cross-streaked plate was photographed for bacteria-killing activity on the following day.

### 2.9. Plasmid Persistence Assay

In order to test the ability of the RP4 conjugative plasmid to persist long-term in its bacterial host, we streaked donor *E. coli* LM715-1 (mScarlet *cam*R, *kan*R, and RP4::GFP *amp*R *tet*R *kan*R) from the frozen stock onto Luria agar plates containing ampicillin (50 μg/mL) and chloramphenicol (20 μg/mL) and incubated the plates aerobically overnight at 37 °C. A single colony was harvested and inoculated on the following day into 3 mL of Luria broth containing ampicillin (50 μg/mL) and chloramphenicol (20 μg/mL) and incubated aerobically at 37 °C for 18 h in a shaker at 150 rpm. Next, we inoculated 30 μL of this overnight culture into each of 8 tubes containing 3 mL Luria broth; 4 tubes had no antibiotics, and the other 4 tubes contained ampicillin (50 μg/mL). These tubes were incubated aerobically overnight at 37 °C in a shaker at 150 rpm. On each of the following ten days, the overnight culture from each tube (30 μL) was transferred into 3 mL fresh Luria broth (1:100 dilution), and ampicillin was added to the respective tubes at 50 μg/mL. The optical density (600 nm) of all cultures was measured every 24 h. We diluted 100 μL of overnight culture from each tube serially in tenfold steps and poured 100 μL of these dilutions on each of three separate Luria agar plates with specific antibiotics on days 1, 2, 3, 5, 7, and 10. Luria agar plates containing chloramphenicol (20 μg/mL) were used to select for total bacteria with and without plasmid, and Luria agar plates containing ampicillin (50 μg/mL) and chloramphenicol (20 μg/mL) were used to select for bacteria-bearing plasmid. These plates were incubated aerobically overnight at 37 °C; colonies were then counted to calculate the proportion of plasmid-bearing cells using the formula $p=\frac{D^{+}}{D^{+}+D^{-}}$ (proportion of plasmid-bearing cells = plasmid-bearing cells (donors)/total bacteria with and without plasmid).

### 2.10. Statistical Analysis

We used an F-test to determine the equality of variances among replicates in the primary and secondary transfers of RP4 plasmid from donor-to-recipient pairings. Based on the results from the F-test, a *t*-test was chosen to calculate statistical significance among treatment groups. An unpaired two-sample *t*-test (independent samples *t*-test) was used to compare statistical differences in mean values of two groups with equal variance. If an unequal variance was observed between groups, Welch’s *t*-test was used to calculate *p* values and significance levels [40]. In all statistical analyses, we used *p* ≤ 0.05 as the cutoff level to determine significance. We also performed statistical analyses after log transformation of the data. However, there were no differences in identifying significant comparisons between groups when compared to using the raw data; that is, no additional significant comparisons were detected, although in most cases, these log transformation analyses produced p values that were lower than 0.05. Thus, all the data and analyses presented here are based on the original data.

## 3. Results

### 3.1. Characterization of Donor and Recipient Strains

To study the conjugation-mediated transfer of ARGs, we developed a set of marked donor and recipient strains to perform traceable in vitro conjugation experiments to document transfer and its frequency. We used a fluorescently labeled commensal donor strain human gut *E. coli* LM715-1 carrying the mScarlet fluorescent protein gene and chloramphenicol and kanamycin antibiotic resistance markers on the bacterial chromosome. A broad-host-range plasmid of RP4 origin carrying green fluorescent protein and ampicillin, kanamycin, and tetracycline antibiotic resistance markers was transferred to the fluorescently labeled donor strain *E. coli* LM715-1. The differential antibiotic and fluorescence markers on the donor bacterial chromosome and plasmid allowed us to select and trace donors, recipients, and transconjugant bacteria using selective media and fluorescence microscopy. Next, we collected a diverse group of strains to test as potential recipients of the RP4 plasmid, including *S.* Typhimurium, *C. rodentium*, *K. pneumoniae*, *V. cholerae*, *P. aeruginosa*, *P. putida*, and three different *E. coli* strains—DEC 5a, LM715-1, and MG1655— that belong to ST-73, ST-259, and K-12 groups, respectively (Table 1). All recipient bacteria also carried unique selection markers not present on the donor bacterial chromosome and RP4 plasmid to determine the selection of genetic markers for the donor, recipient, and transconjugant cells (Table 1). To determine the conjugation frequency of RP4 plasmid among the same strain of *E. coli* LM715-1, we used a donor *E. coli* LM715-1 strain that carried chromosomal mScarlet, along with two antibiotic resistance markers (CamR, and KanR), and the recipient strain, *E. coli* LM715-1, which had a rifampicin resistance marker on its chromosome (Table 1).

We further characterized commensal human gut *E. coli strain* LM715-1 by performing whole-genome sequencing analysis. Based on the Achtman MLST scheme [29], *E. coli* LM715-1 was found to belong to sequence type ST141 lineage B2 and predicted serotype O2:H6. The allele numbers for seven loci are as follows: *adk*, 13; *fumC*, 52; *gyrB*, 10; *icd*, 14; *mdh*, 17; *purA*, 25; and *recA*, *17*. Uropathogenic *E. coli* are often found to belong to ST141 lineage B2 [41], Shiga toxin-producing *E. coli* [42], or hybrid Shiga toxin-producing and uropathogenic *E. coli* [43]. The hybrid position of ST141 *E. coli* strains could serve as a “melting pot” for pathogroup conversion. We also searched metadata of isolates related to ST141 *E. coli* strains in the EnteroBase dataset containing 235,990 *E. coli* strains. Most of those ST141 *E. coli* strains have not been characterized, but those strains that have been characterized are often uropathogenic *E. coli.*

Next, we searched for virulence factors highly associated with Shiga toxin-producing and UPEC *E. coli* bacteria reported in past studies [44,45,46,47]. We did not find *stx* genes that encode Shiga-toxin in the assembled genome of *E. coli* LM715-1. However, we found numerous UPEC virulence genes in the genome of *E. coli* LM715-1, notably *chuA* (heme-binding protein/outer membrane heme receptor), *fyuA* (outer membrane iron receptor), *iroN* (siderophore receptor), *fimH* (mannose-specific adhesin of type I fimbriae), *vat* (Per-activated serine protease autotransporter enterotoxin EspC), and other virulence factors (*sfa*, *pap*, ent, gsp, ybt, and *foc* genes) listed in Appendix A. These findings suggest that commensal human gut *E. coli* LM715-1 is closely associated with uropathogenic *E. coli* strains, which is consistent with our STEC Center-based MLST and Achtman MLST scheme typing methods, which both clustered *E. coli* LM715-1 with UPEC strains.

### 3.2. The Labeled RP4 Plasmid Transferred to Multiple Drug-Resistant Bacterial Strains

To test whether this broad-host-range RP4 plasmid carrying multiple antibiotic-resistant genes can transfer to a diverse set of clinical, commensal, environmental, and opportunistic bacterial strains, we performed in vitro conjugation experiments mating the mouse-adapted commensal human gut *E. coli* LM715-1 plasmid donor with multiple recipient strains. We found that the *E. coli* LM715-1 commensal donor strain successfully transferred the RP4 plasmid to multiple clinically relevant recipient strains (Figure 2), supporting the conclusion that the plasmid can effectively transfer to various bacteria belonging to *Gammaproteobacteria*. However, when we performed a conjugation experiment with *P. aeruginosa*, we found that this recipient killed the donor bacteria during incubation on filter paper for conjugation, suggesting that the *P. aeruginosa* strain carries a bacteriocin or another antibacterial defense mechanism. We were unable to detect donors and transconjugants, even at the lowest serial dilutions of 10^−2^ and 10^−3^, whereas the recipient *P. aeruginosa* was found at ~4.6 × 10^10^ following donor and recipient overnight incubation on filter paper. Further testing by cross streaking the two strains on an agar surface supported this conclusion (Figure 2B).

Next, we tested the hypothesis that the RP4 plasmid transfer frequency would be greater among phylogenetically closely related donor and recipient strains. Phylogenetic relatedness based on 16S rRNA has been studied in *Enterobacteriaceae* (*E. coli*, *C. rodentium*, *S.* Typhimurium, and *K. pneumoniae*) [48,49], *Vibrionaceae* (*V. cholerae*) [50], and *Pseudomonadaceae* (*P. aeruginosa*, *P. putida*) [51]. We found that the transfer frequency of the plasmid was different for each combination of donor *E. coli* LM715-1 and various enteric recipient strains, regardless of phylogenetic relatedness (Figure 2). We performed a statistical analysis using an unpaired independent samples *t-*test to compare the mean values of transfer frequencies from one donor *E. coli* LM715-1 strain to all recipients. The mean self-transfer frequency of donor *E. coli* LM715-1 to recipient *E. coli* LM715-1 was used as a reference value. It was observed that, except for *V. cholerae*, plasmid transfer frequencies were higher among more distantly related bacteria (*K. pneumoniae* and *P. putida*) than closely related bacteria (*E. coli* DEC 5a, *E. coli* MG1655, *C. rodentium*, and *S.* Typhimurium). These findings indicate that the transfer of the RP4 plasmid is not dependent on the phylogenetic relatedness of the donor strain to the recipient strain; thus, we rejected our hypothesis.

Next, we assessed the effect of the antibiotic ampicillin on the transfer frequency of the RP4 plasmid carrying the beta-lactamase gene from the commensal donor strain *E. coli* LM715-1 to different recipient bacterial strains. Based on the antibiotic resistance profiling of recipient bacteria, we divided them into two groups: susceptible to ampicillin (50 μg/mL) and resistant to ampicillin (50 μg/mL). We hypothesized that, first, the presence of ampicillin would increase the transfer frequency among recipient bacteria susceptible to ampicillin and, second, ampicillin would not increase plasmid transfer frequency among those resistant to ampicillin. We used ampicillin (50 μg/mL) for selection in these conjugation experiments. Overall, we found that ampicillin selection pressure increased plasmid transfer among all recipient strains, except *K. pneumoniae* (Figure 3A). For comparison of conjugation in the presence and absence of ampicillin, we performed an unpaired independent samples t-test between no-antibiotic and antibiotic treatment groups. The RP4 plasmid transfer frequencies were significantly increased in ampicillin-susceptible recipient strains *E. coli* MG1655 and *C. rodentium* exposed to ampicillin.

Similarly, a significant increase in the plasmid transfer frequency was observed in ampicillin-resistant *V. cholerae*. However, the presence of ampicillin significantly decreased plasmid transfer to *K. pneumoniae.* We also calculated the donor–recipient ratio for both groups treated with and without ampicillin during conjugation. We determined that antibiotic treatment increased the donor–recipient ratio among all bacteria, except *V. cholerae* (Figure 3B). This outcome suggests that this dose of ampicillin was selecting for carriage of the plasmid in donor and transconjugant cells; however, the influence of the donor–recipient ratio on plasmid transfer frequency was not examined further.

### 3.3. All Recipient Bacterial Strains Can Mediate Secondary Transfer of the BHR Plasmid to the Human Commensal E. coli LM715-1 Recipient Strain

After observing the transfer of the BHR RP4 plasmid from one commensal *E. coli* donor strain to multiple bacterial strains, we tested whether these recipients of the plasmid could transfer it to a commensal *E. coli* strain by acting, in turn, as plasmid donors. Transconjugant bacteria from the previous experiment were used as donor strains, including *S.* Typhimurium, *C. rodentium*, *K. pneumoniae*, *V. cholerae*, *P. putida*, and three different *E. coli* strains. The *E. coli* LM715-1 mScarlet-labeled chloramphenicol-resistant strain was used as a recipient strain in this series of experiments. We performed a statistical analysis using an unpaired independent samples *t*-test to compare the mean values of transfer frequencies individually from each donor strain to the recipient *E. coli* LM715-1 strain. The mean self-transfer frequency of donor *E. coli* LM715-1 to recipient *E. coli* LM715-1 camR was used as a control to provide a reference value. We found that all transconjugant strains transferred the RP4 plasmid to the naïve *E. coli* LM715-1 *cam*R recipient strain at frequencies that ranged from 10^−2^ to 10^−7^ (Figure 4A). These results show that this RP4 plasmid carrying antibiotic resistance genes can efficiently move among multiple bacterial strains in this in vitro model, and once transferred, the original recipients can then serve as plasmid donors for further transfers. After determining the frequencies for these secondary transfers, we also compared plasmid transfer frequencies for each bacterial strain in the primary transfer in which those strains acted as the recipient and in the secondary transfer in which the recipients of primary transfer acted as the donor (Figure 4B). As donors, *E. coli* DEC 5a, *S.* Typhimurium, *P. putida*, and *V. cholerae* showed higher plasmid transfer frequencies than they did as a recipient strain receiving the conjugative RP4 plasmid in the primary transfer. *E. coli* MG1655 showed higher plasmid acquisition in the primary transfer than did *E. coli* LM715-1 when receiving RP4 from *E. coli* MG1655 in the secondary transfer. However, in terms of frequencies, *C. rodentium* and *K. pneumoniae* showed similar abilities to act as a recipient when acquiring RP4 and as a donor when transferring the RP4 plasmid (Figure 4B). These findings show that one must consider how transfer frequencies characteristic of each specific bacterial strain acting in the role of donor or recipient during conjugation affect the efficiency of the spread of the RP4 plasmid.

We were also interested in studying the spread of the RP4 plasmid from *P. putida* and *V. cholerae* to different coliform bacteria other than the donor *E. coli* LM715-1. In these trials, we observed that both donors, *P. putida* and *V. cholerae*, successfully transferred the plasmid to two coliform bacteria, *C. rodentium* and *K. pneumoniae*, in frequencies ranging from 10^−2^ to 10^−7^ (Figure 5).

Furthermore, we performed an in vitro conjugation experiment between donor *E. coli* LM715-1 and mixed coliform fecal bacterial populations freshly grown on MacConkey agar from mice transplanted with a second, distinct infant-derived fecal microbiota. The purpose of this experiment was to detect RP4 plasmid transfer from commensal donor *E. coli* LM715-1 to other gut-resident bacteria. A two-step screening procedure employing the chromosomal and plasmid-borne antibiotic resistance markers was utilized as described in Materials and Methods. The first step consisted of the isolation of a mixed population of donor cells and potential transconjugants on MacConkey agar containing plasmid-borne ampicillin and kanamycin resistance. Of approximately 200 colonies from this population screened in the second step, two were found to be chloramphenicol-sensitive, indicating that they were candidate recipients of the second infant microbiota. RP4 plasmid acquisition in these putative transconjugants was confirmed using colony PCR for the GFP and mScarlet genes and fluorescent microscopy (Figure 6). The transconjugants were characterized using MALDI-TOF mass spectrometry and determined to be *E. coli*. All of these findings of RP4 transfer demonstrate that donor bacteria carrying the RP4 plasmid can effectively spread antibiotic resistance genes to members of the gut microbiota.

### 3.4. Fitness Cost and Persistence of Broad-Host-Range Plasmid during Adaptation and Evolution in a Naïve Host Bacterium

The RP4 plasmid is a large broad-host-range plasmid (~60 Kbp, 2–6 copies of plasmid per cell) and carries self-replication and transmission machinery [13,52]. We tested whether this plasmid would be maintained or lost by a naïve host donor bacterium such as the human gut commensal *E. coli* LM715-1 due to a fitness cost to the host cell. We hypothesized that the RP4 plasmid would persist in a naïve host without antibiotic selection pressure. We used an experimental evolution approach to passage *E. coli* LM715-1 with mScarlet, *cam*R, *kan*R and RP4::GFP *amp*R *tet*R *kan*R markers in fresh growth media daily in the presence and absence of the antibiotic ampicillin (50 μg/mL) for 10 days to test the hypothesis. We calculated the proportion of plasmid-bearing cells (bacteria with plasmid divided by total bacteria in the culture) every 24 h during this host–plasmid evolution and adaptation experiment. The proportion of plasmid-bearing cells declined rapidly to 20 percent on day two post inoculation in cultures with and without antibiotic selection and then decreased slowly throughout the remainder of the 10-day experiment (Figure 7A,B). This result suggests plasmid carriage imposed a fitness cost in this environment, which led to the quick loss of plasmid from the donor strain at the start of the experiment. However, the plasmid-bearing *E. coli* LM715-1 cell population remained persistently present at a low proportion (3–4%) for up to 10 days. Although the proportion of donor bacteria carrying plasmid was lower in the population on days 2–10 of the experiment, a significant number of colony-forming units (1 × 10^8^ cfu) of donor *E. coli* LM715-1 bacteria persisted on day 10 (Figure 6B)—enough to give rise to further plasmid transfer.

## 4. Discussion

Horizontal gene transfer (HGT), which allows bacterial pathogens to acquire resistance genes from other bacteria, has contributed significantly to the spread of antimicrobial resistance among life-threatening pathogens [7,53,54]. Our study has shown that the RP4 plasmid effectively transferred to clinically important and emerging drug-resistant strains of *Enterobacteriaceae*, *Vibrionaceae*, and *Pseudomonadaceae* when transferred by a human-derived commensal *E. coli* donor and tested in the absence of antibiotic selection pressure. Next, we demonstrated that further secondary transmission of the RP4 plasmid to a new recipient occurred and that the primary and secondary transfer frequencies of the RP4 plasmid varied across multiple donor–recipient pairings; these findings are consistent with previous studies [21,22,55]. Thus, the broad-host-range RP4 plasmid is able to replicate and initiate the conjugation process, making it capable of transferring to numerous bacteria and spreading drug resistance, including to bacteria of the human gut microbiota [15,22].

We found that the selection pressure imposed by the presence of the antibiotic ampicillin increased RP4 plasmid transfer in most donor–recipient combinations. However, in the case of *K. pneumoniae*, the conjugation frequency decreased under antibiotic selection with ampicillin. Other studies have reported that neither high nor subinhibitory concentrations of beta-lactam antibiotics increased the plasmid transfer frequency in *K. pneumoniae* and *E. coli* [56,57]. The effect of antibiotic selection pressure on the conjugation rate has also been shown to vary based on donor–recipient pairing, plasmid type, and ecological factors [57,58,59]. We observed that the ratio of donor-to-recipient bacteria increased in the antibiotic-treated group compared to the no-antibiotic group, suggesting that plasmid selection and maintenance in donor bacteria may increase conjugative efficiency. However, the plasmid transfer frequency under antibiotic treatment could be influenced by other factors, such as death of recipient bacteria caused by the drug, the growth rate of transconjugant bacteria, and secondary transfer of the plasmid [57]. All of these factors must be considered when trying to understand transfer frequencies in more complex communities such as the human gut microbiota.

We also found significant differences in a specific bacterial strain’s primary and secondary plasmid transfer frequency depending on whether it was acting as a donor or as a recipient during the conjugation process. It is known that HGT of a conjugative plasmid involves multiple cellular processes [60] and that success of a transfer is determined by genetic and physical characteristics of the plasmid, donor, and recipient strains [61]. For example, each bacterial species has different immunity or defense mechanisms against foreign genetic material. Restriction-modification systems (RMS) [62], anti-phage defense systems [63], and the CRISPR-Cas system [64] are the most well-known bacterial arsenals to prevent the acquisition of foreign DNA and can all alter the transfer frequency of plasmids among bacterial species. Similarly, studies have shown that many bacteria secrete bacteriocins to compete against different species [65,66]. We observed such killing in a conjugation experiment in which a *P. aeruginosa* strain used as a recipient killed the commensal donor *E. coli* LM715-1 strain so that no cells were left to donate plasmid. It is important to note that conjugation efficiency can also vary if the doubling times are different for each donor and recipient bacteria, leading to different donor–recipient ratios during the conjugation experiment [67]. Similarly, other physicochemical factors such as growth phase, growth media, temperature, and mating surface (liquid or solid) can affect conjugation efficiency [56,57,58,59,60,61,62,63,64,65,66,67,68]. In our experiments, we used equal numbers of colony-forming units for each donor and recipient strain tested, but further work is required to determine the effects of dose on the response of these donor–recipient plasmid transfers.

In a serial passage experiment, we found that the fraction of RP4 plasmid-bearing cells rapidly decreased to 20 percent of the population after 48 h; however, afterward, the rate of decline slowed, and the donor bacteria remained in a low proportion throughout the ten-day experiment. Most plasmid adaptation studies have been conducted for more extended periods of 75 days [69,70]. However, another study showed that a much more rapid adaptation of a pKP33 plasmid encoding CTX-M-15 extended spectrum beta-lactamases (ESBLs) and carbapenemases in *E. coli* within a 10-day time-course experiment [71]. We found that it is entirely possible that presence of the RP4 plasmid in a host gut microbiota for even a short period of time may be sufficient to enable further spread of this broad-host-range plasmid to other bacterial species. Critical studies in other environments support this conclusion. For instance, multiple studies have shown that laboratory donor *E. coli* strains carrying conjugative plasmids can spread the plasmid in complex systems such as the gut microbiota in even a few hours [72], as well as in soil and sewage microbiota [11,17,26]. Moreover, it has been shown that multiple conjugative plasmids, including RP4, can persist within host bacteria, even in the absence of antibiotics [73].

One must also consider that conjugative plasmids can adapt to host bacteria by several mechanisms, including by ameliorating fitness costs through compensatory mutations either on the chromosome or plasmid [71,74], carriage if beneficial resistance genes are present [75,76], coevolution under antibiotic selection [77], the presence of multiple copies of plasmid [78], or promotion of a high frequency of plasmid transfer [79]. Conjugative plasmids, especially broad-host-range plasmids, can initiate conjugation and transfer to multiple bacteria without antibiotic selection pressures, as we have shown here. Through continuous conjugation among a network of multiple bacterial cells, such plasmids have another mechanism to persist in host bacterial populations or in complex communities, even with a low level of conjugation frequency [73,80,81]. We also found that the presence of the antibiotic ampicillin did not increase the proportion of plasmid-bearing cells compared to the no-antibiotic group in the conjugation experiments. It has been shown that positive selection of plasmid does not always maintain the plasmid in the host but can also cause plasmid instability and loss [77,82]; however, the primary mechanism behind plasmid instability remains unknown. We used ampicillin, which falls into the beta-lactam class of antibiotics, as our selective drug in this study. It is possible that *E. coli* LM715-1 carrying this plasmid produced a beta-lactamase that degraded the ampicillin, leading to no effect of the antibiotic on overall bacterial growth during the evolution experiment [83,84].

Our study implies that broad-host-range plasmids such as RP4 carrying ARGs are likely to spread antibiotic resistance quickly to multiple species in complex communities, even when under no antibiotic selection pressure. We expect that RP4 plasmids encoding beta-lactamase can maintain themselves in a complex community for more extended periods because they mediate sufficient conjugation frequency for transfer to a diverse range of bacteria to occur [20,22]. The plasmid transfer frequencies generated in this study can be used to build a mathematical model to predict the spread of the conjugative RP4 plasmid across multiple species in a complex environment such as the gut microbiota, where it is difficult to determine actual plasmid transfer frequencies. Such a model could be further extended by adding other clinically relevant plasmids from the IncF, IncI, IncN, and IncQ subgroups [9]. In addition to transfer frequencies, the incompatibility type of broad-host-range plasmids, donor–recipient pairings, and environmental factors related to the conjugation process are also important and should be considered in such a model [24,25,26,27,28,29,30,31,32,33,34,35,36,37,38,39,40,41,42,43,44,45,46,47,48,49,50,51,52,53,54,55,56,57,58,59,60,61,62,63,64,65,66,67,68,69,70,71,72,73,74,75,76].

## Figures and Tables

**Figure 1 microorganisms-11-00193-f001:**
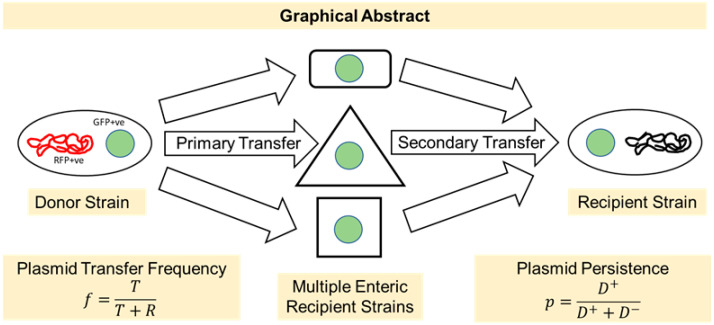
Graphical abstract showing the experimental design used in this study. We studied the transfer frequencies of a broad-host-range RP4 plasmid among multiple clinically relevant bacteria. In the primary transfer, the RP4 plasmid was transferred from *E. coli* LM715-1 to different enteric commensal and pathogenic bacteria, and in the secondary transfer, we observed the transfer of RP4 plasmid to *E. coli* LM715-1 from all plasmid recipients of primary transfer. Plasmid persistence was also studied in *E. coli* LM715-1 using a serial passage approach.

**Figure 2 microorganisms-11-00193-f002:**
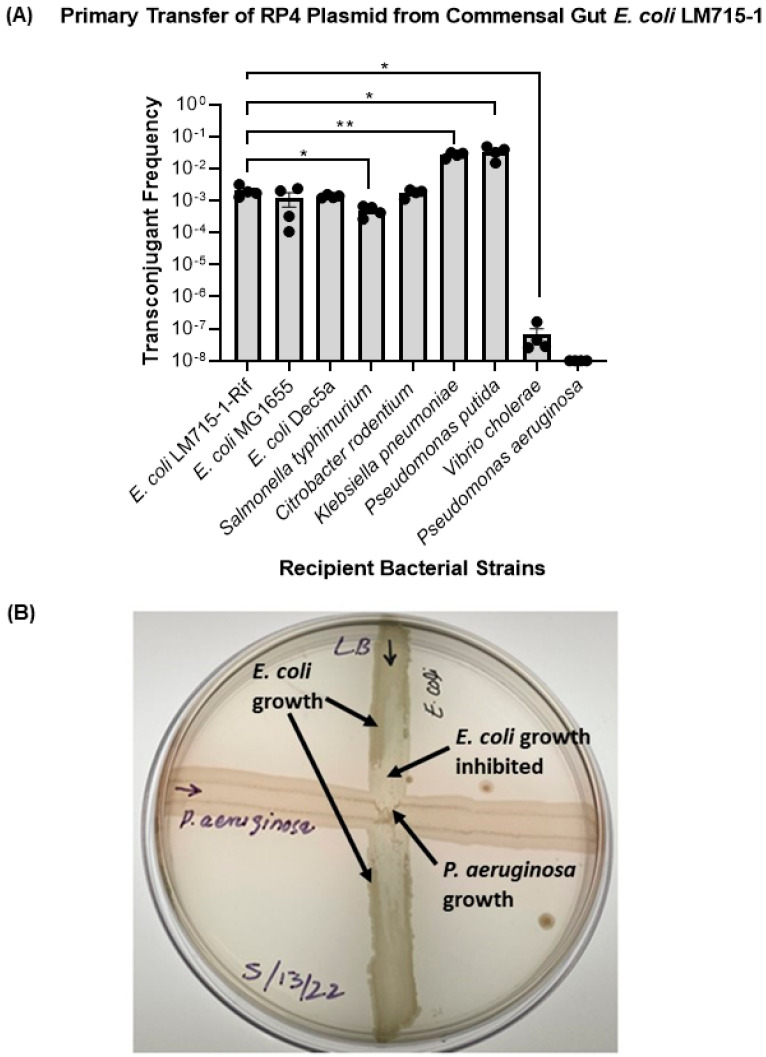
Transfer of RP4 plasmid from commensal strain *E. coli* LM715-1 to multiple recipient bacteria and inhibition by *P. aeruginosa*. (**A**) Plasmid transfer frequencies for all recipients were calculated using the formula (plasmid transfer frequency = transconjugants/(transconjugants + recipients)). Bars show mean values ± standard error of the mean (SEM) based on four independent replicates (n = 4) performed for each conjugation experiment. Plasmid transfer frequencies from *E. coli* LM715-1:RP4 to all other isolates were compared to the plasmid transfer frequency from *E. coli* 715-1:RP4 to *E. coli* LM715-1 RifR. We used Welch’s *t*-test instead of an unpaired independent samples *t-*test if there was unequal variance between compared groups to calculate *p* values. Only significant *p* values are annotated in the graph: * = *p* ≤ 0.05, ** = *p* ≤ 0.01. The transfer frequency was not detected for the conjugation experiment of donor strain *E. coli* LM715-1 with *P. aeruginosa*. (**B**) Growth-inhibitory activity of *P. aeruginosa* against commensal strain *E. coli* LM715-1; zones of growth and inhibition are indicated by arrows. *E. coli* LM715-1 diminishes close to the intersection compared to the corners of the streak, whereas *P. aeruginosa* growth remains intact.

**Figure 3 microorganisms-11-00193-f003:**
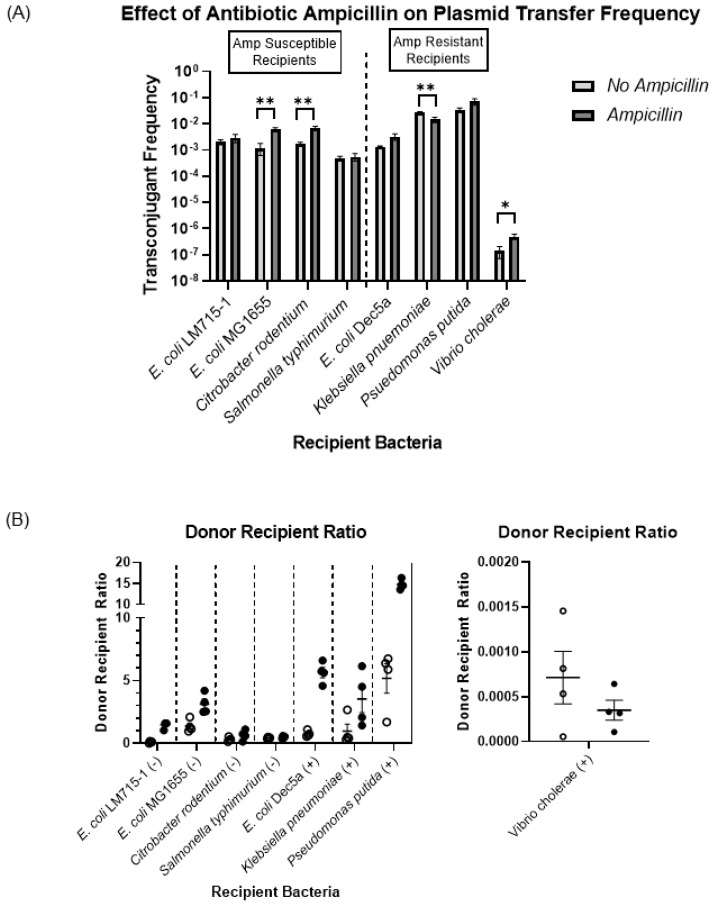
Effect of ampicillin on plasmid transfer frequency. (**A**) Plasmid transfer frequencies for all recipients as calculated using the formula (plasmid transfer frequency = transconjugants/(transconjugants + recipients)). Bars show mean values ± standard error of the mean (SEM) based on four independent replicates (n = 4). For each recipient, frequency of transfer from *E. coli* LM715-1:RP4 in the absence of ampicillin was compared to the transfer frequency in the presence of ampicillin. An unpaired independent samples *t-*test was performed to calculate *p* values; only significant *p* values are annotated in the graph: * = *p* ≤ 0.05, ** = *p* ≤ 0.01. (**B**) Donor-to-recipient ratio with and without ampicillin. The ratio was calculated using the total number of donor and recipient bacteria after 24 h of incubation on filter paper for conjugation on Luria agar plates with and without ampicillin. Circles show four independent replicates (n = 4) performed for each ampicillin-treated (filled circles) and untreated (empty circles) group. Bars show mean values ± standard error of the mean (SEM) based on replicates in each group. A negative symbol (-) in parenthesis indicates susceptibility to ampicillin, and a positive sign (+) indicates resistance to ampicillin. The transconjugant frequency shown in (**A**) was computed using the same number of recipient bacteria used to calculate the donor-to-recipient ratio.

**Figure 4 microorganisms-11-00193-f004:**
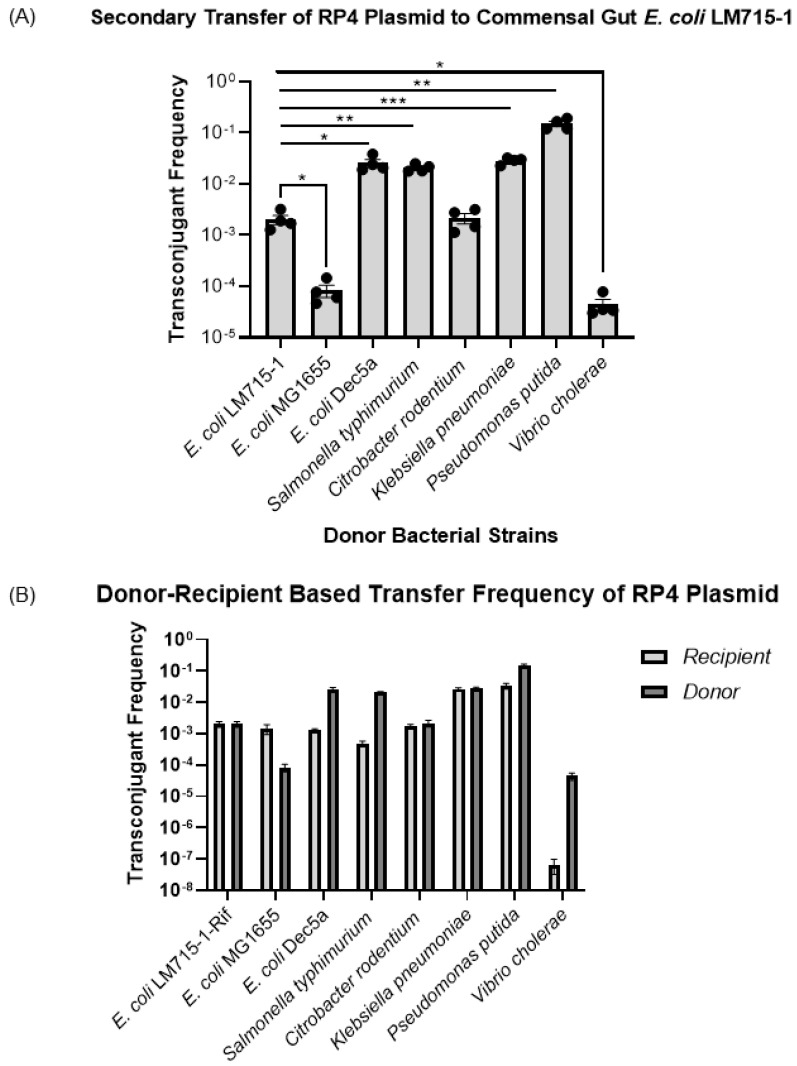
Transfer frequencies of RP4 plasmid to *E. coli* LM715-1 from different donor strains. (**A**) Secondary transfer frequencies of the BHR RP4 plasmid from different recipient bacterial strains shown in Figure 3 back to the commensal recipient *E. coli* LM715-1 strain. The plasmid transfer frequencies for all recipients were calculated using the formula (plasmid transfer frequency = transconjugants/(transconjugants + recipients)). Bars indicate mean values ± standard error of the mean (SEM) based on four independent replicates (n = 4) performed for each conjugation experiment. For each donor, frequency of transfer to *E. coli* LM715-1 was compared to transfer frequency from *E. coli* LM715-1:RP4 to *E. coli* LM715-1 RifR. We used Welch’s *t*-test instead of an unpaired independent samples *t-*test if there were unequal variances between compared groups; only significant *p* values are annotated in the graph: * = *p* ≤ 0.05, ** = *p* ≤ 0.01, *** = *p* ≤ 0.001. (**B**) Transfer frequencies for each strain in the primary transfer (when acting as recipient) and the secondary transfer (when acting as donor) of the RP4 plasmid.

**Figure 5 microorganisms-11-00193-f005:**
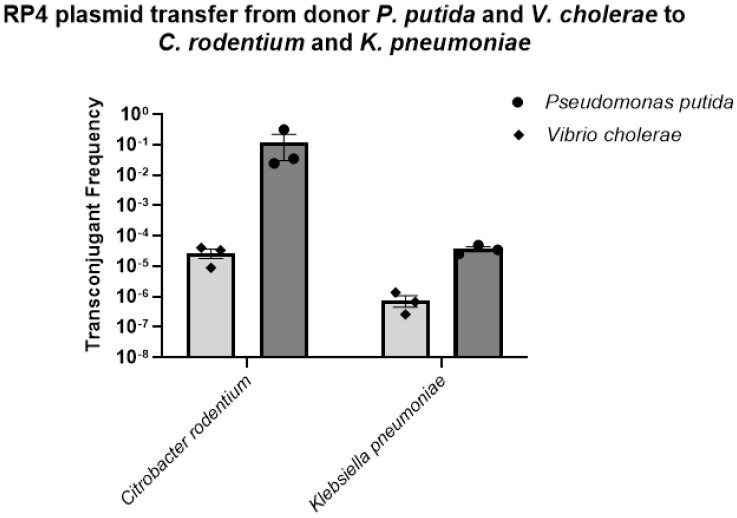
Transfer frequencies of the BHR RP4 plasmid from donors *Pseudomonas putida* and *Vibrio cholerae* to recipient coliform bacteria *Citrobacter rodentium* and *Klebsiella pneumoniae*. The plasmid transfer frequencies for all recipients were calculated using the formula (plasmid transfer frequency = transconjugants/(transconjugants + recipients)). Bars show mean values ± standard error of the mean (SEM) based on three independent replicates (n = 3) performed for each conjugation experiment.

**Figure 6 microorganisms-11-00193-f006:**
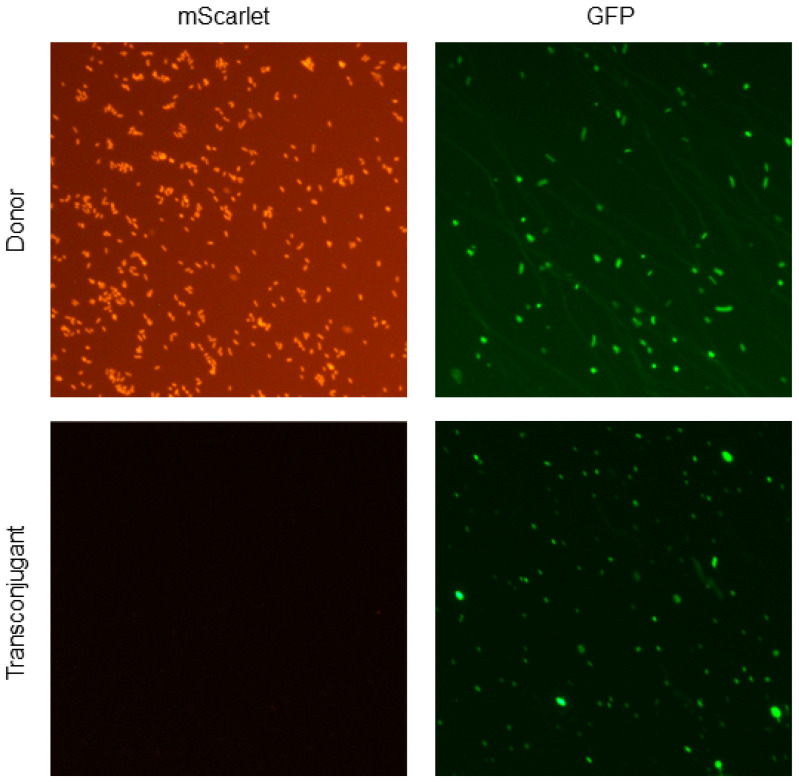
Fluorescent microscopic confirmation of RP4 plasmid transfer from the donor *E. coli* LM715-1 to transconjugant bacteria. The upper row of micrographs in Figure 6 shows that donor *E. coli* LM715-1 bacteria express both chromosomal mScarlet and plasmid-encoded GFP. The lower row shows the transconjugant bacteria expressing only plasmid-encoded GFP. All micrographs were taken using a Nikon Eclipse N*i*-U upright microscope (Nikon, Tokyo, Japan) equipped with GFP and RFP filters at 20× magnification.

**Figure 7 microorganisms-11-00193-f007:**
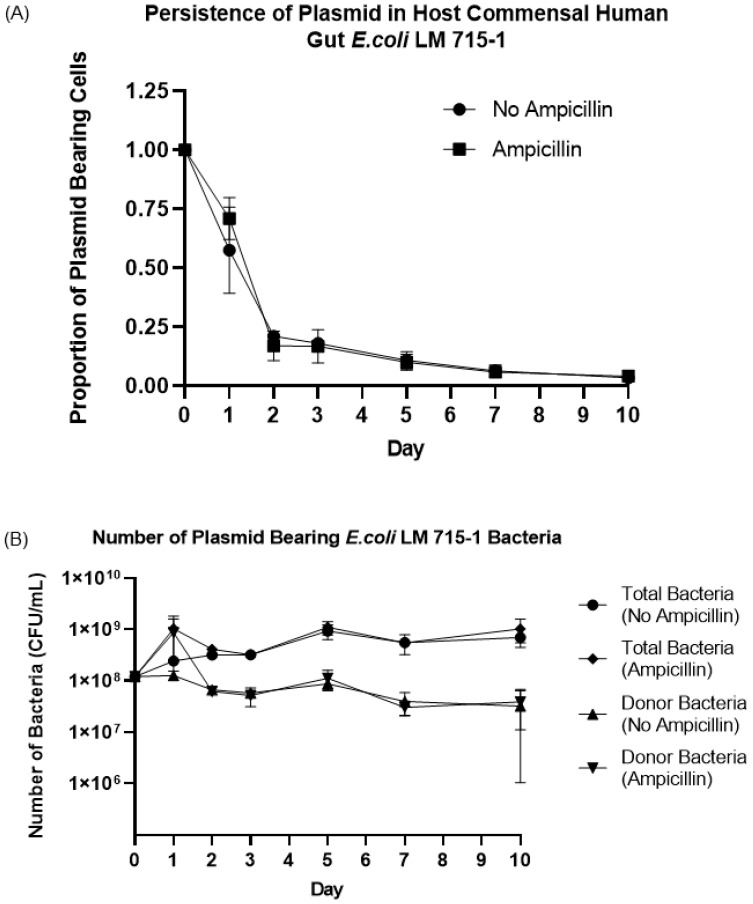
Proportion and number of donor bacteria carrying the RP4 plasmid during serial passage in medium with or without ampicillin. (**A**) Proportions of bacterial cells carrying RP4 plasmid (donor bacteria) and total bacteria (with and without plasmid) were calculated in both groups (ampicillin-treated and untreated) on days 0, 1, 2, 3, 5, 7, and 10. Bars indicate mean values ± standard error of the mean (SEM) based on three independent replicates (n = 3) of each experimental group. (**B**) The actual number of bacteria carrying plasmid and total bacteria with and without plasmid is shown for each treatment group. Bars indicate mean values ± standard error of the mean (SEM) based on three independent replicates (n = 3) of each group of the experiment.

**Table 1 microorganisms-11-00193-t001:** Bacterial strains with their antibiotic resistance profiles and isolation sources.

Description of Donor and Recipient Strains Included in This Study			
Bacterial Strain	Antibiotic Resistance Phenotype	Isolate Type	Source (Attached References)
*Escherichia coli* LM715-1		Human *E. coli* Strain—ST 259	Linda S. Mansfield lab (This study)
*Escherichia coli* LM715-1	CamR, KanR	Human *E. coli* Strain—ST 259	Linda S. Mansfield lab (This study)
*Escherichia coli* LM715-1	RifR	Human *E. coli* Strain—ST 259	Linda S. Mansfield lab (This study)
*Escherichia coli* MG1655	RifR	K-12 *E. coli* Laboratory Strain	ATCC/Lixin Zhang lab
*Escherichia coli* DEC 5a TW00587	AmpR, RifR	Human Diarrheagenic *E. coli* Strain (DEC)—ST 73	STEC/Shannon Manning lab [31]
*Citrobacter rodentium* ATCC 51459	RifR	Pathogen Strain	ATCC/Linda S. Mansfield lab
*Pseudomonas putida* KT2440	AmpR, CtxR, RifR	Environmental Strain	ATCC/Lixin Zhang lab
*Pseudomonas aeruginosa*	AmpR, CtxR, CamR	Human Pathogen Strain Isolated from CF Patient	Robert Quin Lab (personal communication)
*Klebsiella pneumoniae* IA565	AmpR, RifR	Human Pathogen Strain	Christopher Waters lab [32,33]
*Salmonella enterica* serovar Typhimurium	RifR	Clinical Strain Isolated from Chicken	Srinand Sreevatsan lab [34]
*Vibrio cholerae* O1 biotype El Tor C6706str2	AmpR, StrepR	Human Pathogen Strain	Christopher Waters lab [35]

CamR, chloramphenicol; KanR, kanamycin; AmpR, ampicillin; CtxR, ceftriaxone; RifR, rifampicin; StrepR, streptomycin; UPEC, uropathogenic *E. coli*; *DEC*, Diarrheagenic *E. coli*; *ST*, *sequence type*; *CF*, *cystic fibrosis*.

**Table 2 microorganisms-11-00193-t002:** Primers used the detection of fluorescent markers in the donor and recipient strains.

Primers Used for Donor and Transconjugant Confirmation				
Primer	Product Size (in bp)	Primer Sequence (5′-3′)	Target Gene	Gene Bank Accession No. References
*gfp*F	182	GGTGAAGGTGAAGGTGATGC	*gfp*	U73901.1
*gfp*R		CTTCTGGCATGGCAGACTTG		
*mScarlet*F	371	CGCGTGATGAACTTTGAAGA	*mScarlet-I*	KY021424.1
*mScarlet*R		TCGCTGCGTTCATACTGTTC		

## Data Availability

The data presented in this study are openly available at the Dryad database with this unique identifier number (doi:10.5061/dryad.v41ns1s16). The data can also be accessed by using this direct link: https://datadryad.org/stash/share/RoV4ZTGX2GXJQhEJ0s4S_jheIn5G0jMxAIEW7b9NRRk, accessed on 2 December 2022.

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
