# Peer review of "Conjugative RP4 Plasmid-Mediated Transfer of Antibiotic Resistance Genes to Commensal and Multidrug-Resistant Enteric Bacteria In Vitro"

_microorganisms, 2023, doi:10.3390/microorganisms11010193_

Round 1
Reviewer 1 Report
In their article, Sher and colleagues characterize conjugative transfer of RP4 plasmid from a commensal E. coli strain to other Enterobacteriaceae and then from these transconjugant to new recipients. Overall, the authors provided some interesting insights into conjugal plasmid transfer among gut commensal strains, but there were several modest corrections that were needed to improve the readability and interpretation of the paper.
Suggested corrections are below:
Line 111 – “pathogen” should be “pathogenic”
Line 125 indicated ceftriaxone was used, while Line 134 indicated cefotaxime was used, please clarify
Line 253 – uL should be mL; meshing should be another word, possibly mixing
Include negative controls for green autofluorescence and gfp gene in fecal slurries?
Line 370 – insert “that” following MG1655 and add “, respectively” following goups on line 371
Since E. coli LM715-1 was used as both donor and recipient strain, please add the antibiotic resistance profile for the strain that contains the chromosomal mScarlett as well as the parental strain to Table 1.
Line 371-374: As written, LM715-1 is both a donor and recipient in one experimental pairing. It is unclear from Table 1 how there are unique antibiotic selective markers that allows differentiation for this specific pairing. Please clarify.
Line 380: Should “.” after “B2” be a comma? Otherwise, the text after Toval et al reference is a sentence fragment.
Line 389: Do you mean that you did not find stx genes in E. coli LM715-1? Otherwise, the meaning is unclear as at least some E. coli will contain stx.
Line 396: Achtman “MLST” not “MLTS”
Line 408-409: Since T6SS are known in P. aeruginosa, perhaps better to refer to “antibacterial defense mechanism” rather than “toxins”
Figure 2A – rather than plotting transconjugant frequency as 10-8 for P. aeruginosa, denote not detected on graph or in figure legend.
Figure 2B – Growth inhibition is not clear in image shown from cross-streaking experiment. Please use an alternative method to demonstrate inhibition. Perhaps a zone of inhibition experiment would be clearer. Also, plotting the data obtained from recipient strains following incubation with P. aeruginosa on filter in supplemental file would provided further support to your conclusion.
Line 424-439: To provide better support, please reproduce a phylogenetic tree showing relatedness between Enterobacteriaceae. Also, the way the sentences is worded in line 429-430 “…for each donor and recipient combination…” suggests more than one donor was tested in Figure 2. According to your legend, this is not true, E. coli LM715-1 is the only donor tested here. Please revise accordingly.
Line 440-461: What controls did you perform that this ampicillin-dependent increase in transfer frequency was not a technical artefact of your method of calculating transfer frequencies? Transfer frequency = tc/tc+recipients. If recipients are susceptible to ampicillin, shouldn’t the denominator decrease in these experiments, thereby increasing the calculated transfer frequency. Did you see significant differences in recipient numbers following filter incubation? Please provide these numbers in supplemental figure.
Line 538-547 should not be included if data supporting these assertions is not shown.
The lack of selective pressure for plasmid maintenance in the presence of ampicillin is somewhat surprising. Since ampicillin resistance is typically through action of a secreted beta-lactamase that can inactivate ampicillin and benefit cells that don’t encode resistance, did you also test plasmid stability using an antibiotic resistance (e.g., kanR) that acts intracellularly and only benefits host cells that encode resistance?
Line 603-605: Your hypothesis that ampicillin increased transfer frequency through plasmid selection and maintenance is not supported by your plasmid stability data, which shows that the plasmid carriage rate is the same in the presence and absence of ampicillin (Figure 6A). Based on this, your alternative hypotheses seem more plausible.
Author Response
Please find in the attachment.

Reviewer 2 Report
1. Some scientific names of bacteria are not italicized;
2. There are words that pass or lack letters, just review
The article has fluid writing. It is an experimental work of basic
information. I felt the need for the authors to discuss a little more
the clinical application of the work. However, the work is informative
and of great relevance.
Author Response
Please find in the attachment.
